# Construction of Genetic Linkage Map and Mapping QTL Specific to Leaf Anthocyanin Colouration in Mapping Population ‘Allahabad Safeda’ × ‘Purple Guava (Local)’ of Guava (*Psidium guajava* L.)

**DOI:** 10.3390/plants11152014

**Published:** 2022-08-02

**Authors:** Harjot Singh Sohi, Manav Indra Singh Gill, Parveen Chhuneja, Naresh Kumar Arora, Sukhjinder Singh Maan, Jagmohan Singh

**Affiliations:** 1Department of Fruit Science, College of Horticulture and Forestry Punjab Agricultural University, Ludhiana 141004, India; misgill@pau.edu (M.I.S.G.); naresh_arora@pau.edu (N.K.A.); sukhjinder-fs@pau.edu (S.S.M.); 2School of Agricultural Biotechnology, Punjab Agricultural University, Ludhiana 141004, India; pchhuneja@pau.edu; 3Division of Plant Pathology, ICAR-Indian Agricultural Research Institute, New Delhi 110012, India; dhillonjagmohansingh@gmail.com

**Keywords:** linkage map, hybrids, SSR markers, anthocyanin, QTL

## Abstract

In the present investigation, F1 hybrids were developed in guava (*Psidium guajava* L.) by crossing high leaf-anthocyanin reflective-index (*ARI*1) content cultivars purple guava (local) ‘PG’, ‘CISH G-1’ and low leaf-*ARI*1 content cultivar Seedless ‘SL’ with Allahabad Safeda ‘AS’. On the basis of phenotypic observations, high *ARI*1 content was observed in the cross ‘AS’ × ‘PG’ (0.214). Further, an SSR-markers-based genetic linkage map was developed from a mapping population of 238 F1 individuals derived from cross ‘AS’ × ‘PG’. The linkage map comprised 11 linkage groups (LGs), spanning 1601.7 cM with an average marker interval distance of 29.61 cM between adjacent markers. Five anthocyanin-content related gene-specific markers from apple were tested for parental polymorphism in the genotypes ‘AS’ and ‘PG’. Subsequently, a marker, viz., ‘*MdMYB*10F1′, revealed a strong association with leaf anthocyanin content in the guava mapping population. QTL (*qARI-6-1*) on LG6 explains much of the variation (PVE = 11.51% with LOD = 4.67) in levels of leaf anthocyanin colouration. This is the first report of amplification/utilization of apple anthocyanin-related genes in guava. The genotypic data generated from the genetic map can be further exploited in future for the enrichment of linkage maps and for identification of complex quantitative trait loci (QTLs) governing economically important fruit quality traits in guava.

## 1. Introduction

Guava (*Psidium guajava* L.), (2*n* = 22) belongs to the family Myrtaceae, is popularly known as ‘Apple of Tropics’ and is also referred to as a super fruit due to its high nutraceutical and medicinal properties [1,2]. Skin colour of fruit is one of the important traits in terms of aesthetic and commercial value and is mainly determined by the composition and availability of anthocyanin content [3,4]. Coloured guava pulp is a rich source of natural antioxidants such as β-carotene and lycopene [5]. Three anthocyanins (cyanidin chloride, malvidin 3-glycoside and cyanidin 3-glycoside) are present in coloured guava pulp [6]. Coloured-skin guava cultivars, which are good source of dietary fibre, have high anthocyanin reflective index and carotenoid content. Coloured skin/flesh guava is highly attractive and demanded by health-conscious people [7]. The major breeding objective of guava is to develop high yielding cultivars with good quality traits such as medium size, soft seeded, coloured skin and fruit pulp [8].

The long juvenile period and the polygenic nature of guava fruit trees are major bottlenecks for conventional breeding programs [9]. Traditional methods assisted with molecular markers and genetic maps offer an alternative approach to speed up the breeding process pertaining to the long juvenile phase, and advanced selections for fruit-related traits at the pre-bearing phase [10,11]. In this regard, a double pseudo-testcross strategy is employed for development of linkage maps in highly heterozygous species of woody perennials [12,13]. The concept of the double pseudo-testcross strategy for development of linkage maps in the F1 generation of fruit crops can be simulated to that of F2 or BC1 of an annual self-fertilizing crop [14]. Linkage maps of progenies segregating for highly contrasting phenotypic traits are required to develop marker-assisted selection to accelerate the genetic improvement of guava.

Limited availability of guava genomic resources is another issue for the use of molecular analysis/profiling techniques in guava breeding/improvement programmes [15,16]. The first molecular linkage map in guava (‘Enana Roja Cubana’ × ‘N6′) was constructed with AFLP markers [17] which were further extended with Amplified Fragment Length Polymorphism (AFLP) and Simple Sequence Repeats (SSR) markers [18]. Later on, guava-specific SSRs were used for increasing the density of an AFLP linkage map [19]. Guava is still considered an orphan crop with reference to its exploration at the genomic level. Only a few reports are available on molecular mining of the guava genome for its quantitative genetics. In addition, QTLs for morphogenic and fruit traits have been integrated into the AFLP maps of an MP1 population of guava [17,18]. Microsatellites or SSRs have been widely used as efficient tools for germplasm characterization, and for diversity studies on *Psidium* germplasm [20,21]. Two separate molecular linkage maps of F1 population (‘Kamsari’ and ‘Purple Local’) of guava, using SSR markers in combination with Sequence-related amplified polymorphism (SRAP) markers, were constructed for identifying the complex QTLs associated with fruit quality traits [22]. The PCR (Polymerase Chain Reaction)-based molecular markers, viz., SSRs or microsatellites, have been widely used for genetic mapping studies in horticultural perennials [23,24,25]. SSRs are co-dominant markers and have gained more importance in plant genetics studies due to their properties of higher polymorphism, abundance, multi-allelic nature, and extensive genome coverage [26,27,28,29]. Previously, SSR markers have also been exploited for molecular discrimination of guava germplasm [9,16,30,31]. White flesh and green/pale-yellow skin colour cultivar Allahabad Safeda, white flesh and apple skin colour cultivator CISHG-1, and purple flesh/dark green to purple skin colour cultivar Purple Guava are widely used in breeding programs in India [32]. Eighty SSR markers were used to study the diversity available in Malaysian Guava (Purple Guava) and Allahabad Safeda. High dissimilarity was found in Allahabad Safed and Purple guava based on the morphological performance and features [33].

To date, there is no report available on the amplification of anthocyanin gene specific markers in guava, and mapping of QTL regions responsible for leaf anthocyanin colouration. Keeping in mind the above facts, the present investigation aims to construct a genetic linkage map in F1 individuals of ‘AS’ × ‘PG’. In order to identity their futuristic corroboration with anthocyanin present in guava leaves and to elucidate whether it is translated into fruit skin colour of guava, candidate gene markers of apple were tested.

## 2. Results

### 2.1. Leaf Anthocyanin Reflective Index (ARI1)

Young, flushed leaves of white fleshed commercial cultivar ‘Allahabad Safeda’ (AS), cv. Seedless (SL), skin coloured cv. CISH G-1, coloured skin with pink fleshed cv. purple guava (PG) and their different cross combinations, viz., ‘AS × PG’, ‘AS’ × ‘CISH G-1′, ‘AS’ × ‘SL’ were surveyed for Anthocyanin Reflective Index (*ARI*1). Parent purple guava (local) ‘PG’ show bright purple colouration on younger leaves followed by light pink colour in the parent ‘CISH G-1′ and green colouration in cultivar Allahabad Safeda ‘AS’ and Seedless ‘SL’ guava leaves (Appendix A). In F1 individuals of ‘AS × PG’ a visual major morphological gene flow (purple colouration) was seen on younger leaves and it changed to light purple colouration with leaf maturity [Figure 1a]. The maximum anthocyanin reflective index (*ARI*1) (Figure 2) was found in parent PG (0.266) (Appendix A). The *ARI*1 content of F1 population (‘AS × PG’) was closest to the male parent PG (0.214). This is suggestive of the movement of colour-changing pigment traits from coloured genotypes in hybridization programmes. Similarly, a visual light pink colouration showed the movement of colour-changing pigment traits [Figure 1b] from male guava parents ‘CISH G-1′. F1 individuals of ‘AS x ‘CISH G-1′ have *ARI*1 content of 0.125, which is closest to *ARI*1 content of parent ‘CISH G-1′ (0.172). Both the parents, ‘PG’ and ‘CISH G-1′, were considered in the present experiment due to their higher *ARI*1 content and parent ‘SL’ with lower *ARI*1 content and presenting green colouration was a check [Figure 1c]. Minimum anthocyanin reflective index of 0.013 with green colouration was observed on younger and mature leaves of parent ‘SL’ and their F1 individuals ‘AS’ × ‘SL’ [Figure 1c]. F1 individuals obtained from ‘AS’ × ‘PG’ had higher *ARI*1 content compared to the F1 hybrids of ‘AS’ × ‘CISH G-1′and ‘AS’ × ‘SL’.

### 2.2. Selection of Mapping Population

Guava parent ‘AS’ with visual green coloured leaf has a lower *ARI*1 content (0.053) and parent ‘PG’ with purple coloured leaf (Figure 1) has the highest *ARI*1 content (0.266) (Table 1). The F1 population was phenotyped for leaf-based colouration traits for the establishment of phenotypic-trait relationship purpose in the future. Five anthocyanin-gene-specific markers were selected {*MdMYB*9; *MdMYB*10; *MdMYB*10 (F1, R1); *MdMYB*10 (F2, R2); *MdMYB*17} (Appendix A). These anthocyanin-gene markers were earlier used in apple for pre-selection of anthocyanin traits at seedling stage. These candidate gene markers were also tested on segregating F1 progenies of ‘AS’ × ‘PG’, ‘AS’ × ‘CISH G-1′, and ‘AS’ × ‘SL’ for amplification. Candidate gene marker *MdMYB*10F1 showed polymorphism between parental lines ‘AS’ × ‘PG’. However, it does not show any polymorphism on any F1 individuals of ‘AS’ × ‘CISH G-1′ and ‘AS’ × ‘SL’. The candidate gene marker *MdMYB*10F1, however showed a differential amplification pattern compared with individuals of ‘AS’ × ‘PG’. A total of 1255 band scores were generated with an average of two-to-four alleles per plant and amplification product ranging from 680 to 850 bp. With regard to pollen parent (PG), the amplification product ranged from 780 to 850 bp. Out of 400 F1 individuals, 238 F1 plants showing Purple-Guava-specific alleles (monomorphism) were selected for the construction of a linkage map and the rest of 162 F1 plants, having female-parent-specific alleles, were discarded.

### 2.3. Generation of Linkage Map

The genomic DNA of 238 F1 individual of ‘AS’ × ‘PG’ which showed high anthocyanin content (*ARI*1) on their leaves (Appendix A) were used as a mapping population. A total of 106 SSR-primer pairs were selected from already published literature, detailed in Appendix A. An initial parental survey of ‘AS’ and ‘PG’ revealed 73.58% polymorphism (i.e., 78 out of 106 SSR primer pairs). Some of these markers were detected at multiple loci in the genome. Genotyping of the mapping plant population resulted an average of 771 alleles, with a maximum of four alleles and minimum of three alleles per primer. Out of these, 482 (62.51%) were found specific to the female parent ‘AS’, and 270 alleles (37.48%) were specific to the male parent ‘PG’ showing a 3:1 allele segregation ratio; the remaining 19 alleles (2.46%) were common to both the genotypes (Appendix A). From these 78 SSR markers, only 69 SSR markers were mapped in the constructed linkage map of F1 individuals ‘AS’ × ‘PG’. Nine SSR markers showed a distorted segregation, accounting for 8.49%. A linkage map consisting 69 markers anchored to 11 linkage groups (LGs)/chromosome (Chr) with total genome coverage of 1601.17 cM (Table 1). A maximum of 11 SSR markers were ordered at Linkage group 6 (LG6). Seven linkage groups (LG1, LG3, LG4, LG7, LG8, LG9 and LG10) consisted of only five SSR markers. The average length of LGs was 145.56 cM and average interval spacing on 11 LGs was 29.61 cM. Considering the relative loci distances, marker mPgCIR22 and mPgCIR47 had the highest distance (60.95 cM) anchored onto LG3. The lowest genetic distance, 1.65 cM, was recorded between markers mPgCIR23 and mPgCIR32. Only those markers that could be ordered at an LOD score of >3.0 were directly included in the linkage-map framework, while the rest of markers were removed from analysis.

### 2.4. QTL Mapping

The phenotypic variation of anthocyanin trait is presented in Figure 3. The two parental lines differed markedly for anthocyanin reflective index (AS = 0.053, PG = 0.266). The population range of anthocyanin reflective index was from 0.0161 to 0.284 with a mean, standard deviation (SD), and coefficient of variation (CV) of 0.214, 0.025, and 11.91%, respectively. The data distribution of the population is somewhat positively skewed (0.20) with a kurtosis value of −0.28 (Appendix A). Transgressive segregation was observed, and this trait showed continuous segregation and is suggested to be under the control of multiple genes. Thus is shown the suitability of the studied trait for QTL analysis. QTL analysis was carried out with the help of QTLIci mapping software using 69 markers of the genetic linkage maps (Figure 4). A total of three putative QTLs for ARI1, viz., qARI-5-1, qARI-6-1, and qARI-8-1 were detected, with each one explaining between 2.56% and 11.51% of the phenotypic variance (Table 2). These QTLs were distributed on three linkage groups, such as LG5, LG6 and LG8; one major effective QTL (qARI-6-1), explaining more than 11.51% PVE, was detected on LG6 with an LOD score = 4.67, and the remaining two minor QTLs (qARI-5-1 and qARI-8-1) were detected on LG5, and LG8, respectively. The additive effect of the main effective QTLs is positive, and the others showed a negative additive effect, with increasing effects of QTLs.

## 3. Discussion

### 3.1. Leaf-Colouration ARI1 Analysis

Quantitatively, guava parent ‘PG’ has the highest Anthocyanin Reflective Index (*ARI*1) content (0.266) and this was closely followed by their F1 hybrids, having *ARI*1 content of 0.214. The *ARI*1 content of F1 hybrids of ‘AS’ × ‘CISH-G1′ was closest to that of the male parent ‘CISH G-1′, whose *ARI*1 content was 0.172 (Figure 3). The lowest *ARI*1 content was recorded in parent ‘SL’ (0.013) and their derived progeny ‘AS’ × ‘SL’ (0.027). This clearly depicted the movement of colour-changing pigment traits during the crossing of guava crop and there was no sign of colouration in green coloured guava cultivars. The newly developed F1 hybrids from both cross combinations ‘AS’ × ‘PG’ and ‘AS’ × ‘CISH G-1′ exhibited morphological changes and reflected a purple and light pink colouration on their younger leaves [Figure 1a,b], whereas there was a green colouration on the younger leaves of cultivar ‘SL’ and their F1 progeny ‘AS’ × ‘SL’ [Figure 1c]. These results are in keeping with a view of complicated morphological properties and the fact that our understanding of the contribution of all the pigment colouration is not yet sufficient. However, considerable progress was made and non-destructive techniques for better assessment of pigments in plant leaves [34,35,36,37,38,39,40] were developed. In earlier studies, the (*ARI*1 content) anthocyanin absorption induced a dramatic decrease in reflectance in the green region of the spectrum [41]. All the guava parents (AS, PG, SL, CISH G-1) and their F1 populations were provided with equal conditions such as light, irrigation and other capital needs which could affect them during this study programme.

### 3.2. Amplification of Candidate Gene Marker

The present study is the third report of SSR-based linkage map development and first of its kind for amplification of anthocyanin-related candidate gene-specific markers in guava. Five candidate gene markers {*MdMYB*9; *MdMYB*10; *MdMYB*10 (F1, R1); *MdMYB*10 (F2, R2); *MdMYB*17}, which were found to be associated with red and non-red skin colour of apple cultivars [42], were applied in guava as pre-selection indices for fruit skin colour in a progeny population of 238 seedlings from the cross between ‘Allahabad Safeda’ and ‘Purple Guava (local)’. In earlier reports, allele-specific markers (*MdMYB*1) were potentially used for accelerating the breeding selections of anthocyanin fruit skin colouration at seedling stage in apple [43,44,45]; these reports demonstrated the use of allele-specific DNA markers for predication of future fruit colour at the beginning of the breeding programs. Most of the previous studies in guava are related to mapping of genomic regions controlling morphogenic and fruit-related traits with RAPD, SSR and SRAP markers [18]. Our approach is based on the amplification pattern scoring of candidate gene marker *MdMYB*10F1 polymorphism on F1 individuals (‘AS × PG’), those that phenotypically exhibited a purple leaf colouration, as well as their quantitative measurements such as *ARI*1. However, it was identified that a candidate gene marker *MdMYB*10F1 only showed polymorphism in response to parents ‘AS’ and ‘PG’ and their F1 progenies, with ‘AS’ × ‘PG’ having a higher value for *ARI*1 content. This result, in combination with previous observations of up-regulation of the anthocyanin gene marker *MdMYB*10 allele, which was found responsible for the elevated accumulation of anthocyanin in red-fleshed apples [45], strongly indicates that *MdMYB*10F1 could be one of genes that control anthocyanin colouration in leaves and it might be translated or reciprocal in fruit skin colour of guava fruits.

### 3.3. Linkage Mapping of Guava

An SSR marker-based genetic linkage map was constructed from one of the guava parents with green coloured leaf, Allahabad Safeda ‘AS’, and another parent with purple coloured leaf, purple guava (local) ‘PG’ (Figure 4). Despite great advances in genomics of horticultural crops, guava has still received meagre attention with respect to molecular profiling. In guava, very few linkage-mapping studies have been reported using a combination of AFLP, RAPD, SRAP and SSR markers [17,18,19,46]. The first linkage map of guava using SSR markers was developed [18] for enrichment of an AFLP-based linkage map of three guava populations [17]. SSR markers are still preferable for linkage mapping studies in horticultural perennials due to their genomic abundance, co-dominant nature, and requirement of basic facilities. A pseudo-test cross strategy is considered to be ideal for development of linkage maps of F1 populations of highly heterozygous fruit crops. The present genetic map was constructed using 69 SSR markers consisting of 11 linkage groups (LGs), with a genome spanning 1601.17 cM and an average distance of 23.20 cM between markers. The length of individual linkage groups was 39.86~358.30 cM, and there were 5~11 markers on different linkage groups. Similar results to our study were reported by [18] using a similar set of 73 SSR markers and LOD score set = 3.0, in which nine LGs covered 1641.75 cM. Whereas, [15] reported an enriched linkage map of guava cv. Kamsari and Purple (local) with 351 SSR markers grouped into 11 LGs spanning a length of 1951.9 cM. An integrated linkage map of guava which contained 578 markers (452 AFLPs, 126 SSRs), distributed on 11 linkage groups, covered a length of 2179 cM. However, discrepancies in marker order may not always represent structural changes in the chromosomes. Some markers can detect multiple loci in the genome and sometimes the difference is due to different LOD values used in different studies and earlier reports showing the same pattern of SSR markers. Altering of LOD values changes probability level; the lower the LOD level, the more markers will be assigned to linkage groups and vice versa. Additional mapping of markers in these regions is required to assign a proper order and get evidence for any structural change.

### 3.4. Mapping of Leaf Anthocyanin Colouration

Furthermore, a genetic linkage map was used to identify quantitative trait loci (QTLs) affecting anthocyanin-associated traits. Three QTLs (*qARI*-5-1, *qARI*-6-1, *qARI*-8-1) for anthocyanin reflective index (*ARI*1) belonged to 3 different linkage groups and explained 2.56–11.51% region (*qARI*-6-1) gathered at the region of 273.5–286.5 cM of LG6 explained 11.51% (LOD = 4.67) flanking between mPgCIR265 and mPgCIR343. One QTL (*qARI*-5-1) was positioned on LG5, with an LOD value of 5.06 and the contribution rate was 4.51%, whereas, *qARI*-8-1 was positioned at LG8, and the contribution rate was 3.44%. The accurate phenotypic assaying is of the utmost importance for the accuracy QTL mapping. A reliable QTL map can only be produced from reliable replicated phenotypic data, which improve their precision by reducing experimental error and background noise. To determine the significance of a QTL, an LOD (log of odds) score ≤ 3 (a ratio of likelihoods of 1000 to 1) is generally used as a critical value. An LOD threshold level range of 2.5–3 is often used as a criterion to declare the QTL significance, and to minimize the error frequency [47]. In addition, the identified QTL regions are mostly described as minor or major. This criterion is based on the PV (phenotypic variation) proportion explained by a QTL (on the basis of r2 value): a major QTL will account for a relatively large amount (≥10%), and a minor QTL will usually account for <10 [48]. Studies on the underlying mechanism of quantitative genetics in guava are meagre due to the complexity involved in generating mapping populations, long juvenility, high heterozygosity, and lack of adequate genomic resources. There are very few reports regarding mapping of genomic regions responsible for morphogenic and physio-chemical fruit traits of guava [17,19,22,46]. To the best of our knowledge, this is the first report on leaf anthocyanin-related QTL identification in guava. The future potentiality of this map for genetic improvement of guava is the enrichment of linkage maps, which can be further used for the fine mapping of the gene of interest (ARI1), gene tagging, MAS breeding and comparative genomics [27]. Earlier, in similar research with less variation for grain protein content (GPC) across environments, the QTLs which control the stability of GPC were identified [49,50].

## 4. Materials and Methods

### 4.1. Genotypes and Study Area

Four different guava cultivars showing contrasting phenotypic characters for colour of leaf, fruit flesh and skin were selected as parents for hybridization. Different crosses were made involving commercial varieties of Punjab (India), viz., Allahabad Safeda (AS) was taken as the female parent and purple guava (local) (PG), CISH G-1 and Seedless (SL) as male parents. Commercial cultivar, AS has fruit with round shape, soft seeds, cream coloured flesh, light green skin colour, and green leaf colour. While PG has fruit with round shape, soft seeds, purple coloured flesh and skin, and having leaf of purple colour. CISH G-1 has fruit with round shape, soft seeds, cream coloured fleshed fruit with red skin, and younger leaf with light pink colour and mature leaf with green colour. Non-commercial cultivar, SL has fruit with non-uniform shaped, cream coloured flesh, and light green skin coloured fruit and leaf.

The hybridization experiments were carried out on young, healthy trees having stability in fruit production and being free from insect pests, maintained in a field at Fruit Research Farm, Department of Fruit Science, Punjab Agricultural University, Ludhiana, Punjab, India. The flowers were emasculated, bagged and hand-pollinated the next morning for three cross combinations, viz., cross combination (I): Allahabad Safeda × purple guava (local) (Appendix A); (II): Allahabad Safeda × CISH G-1; and (III) Allahabad Sageda × Seedless. The hybrid fruits were harvested when fully mature and seeds collected. Collected seeds were stratified in moist, finely ground peat moss in polybags. Seedlings were maintained/raised in individual pots/polybags under shade net house during the years 2015–2017. Since the guava is a cross-pollinated crop and the parental genotype was heterozygous so the hybrid population was expected to segregate for the target traits.

### 4.2. Anthocyanin Reflective Index (ARI1)

Measurements were made non-destructively from outer canopy of ten younger leaves of each of the fruit plants/F1 hybrids. Anthocyanin (*ARI*1) was recorded using a portable Spectra Snap {CI-710 Miniature Leaf Spectrometer (CID Bio-Science, Camas, WA, USA)}. The average of the ten spectral readings was estimated and the averaged signature reading was considered. This index uses reflectance measurements in the visible spectrum to take advantage of the absorption signatures of stress-related pigments [39].
$$\mathrm{ARI}1=\frac{1}{\rho_{550}}-\frac{1}{\rho_{700}}$$

### 4.3. Statistical Analysis

*ARI*1 data was subjected to one-way analysis of variance (ANOVA) using stats package of R (version 3.1.3, https://www.r-project.org, accessed on 29 March 2022) in R v4.0.3 (Core R Team 2019) with *p* values ≤ 0.05 considered as statistically significant and means separated using the least significant difference (LSD) test.

### 4.4. Linkage Mapping and Genomic Analysis

#### 4.4.1. Guava DNA Analysis

The genomic DNA was extracted from fresh young expanded leaves of the 238 F1 hybrids and their respective parents using the modified CTAB method described by [51]. The purity and integrity of isolated DNA was determined by electrophoresis in 0.8% agarose gels. DNA was amplified in vitro through polymerase chain reaction (PCR) using SSR primers following standard protocols [52]. PCR amplification was carried out in 20 µL reaction-mixture containing 10X PCR buffer (10 mM Tris-HCl, pH 8.3, 50 mM KCl, 1.5 mM MgCl_2_, 0.01% Gelatin), 1.5 mM MgCl_2_, 150 µMdNTPs, 0.75 µM forward primer, 0.75 µM reverse primer, 45 ng genomic DNA, and 1 unit of Taq DNA polymerase (Bangalore Genei, India). The reaction was performed in a 96-well microtitre plate in an Eppendorf Master cycler gradient using 15 ng of genomic DNA of each genotype. Finally, the PCR products were separated on an 8% native polyacrylamide gel electrophoresis (PAGE) and stained at 60 V for an initial 3 h with ethidium bromide (0.5 μg/mL) followed by 100 V for another 1 h (Appendix A). A standard molecular weight marker (mass ruler DNA ladder, MBI Fermentas) was used in each electrophoretic run and the UV transilluminated gels were photographed in gel documentation and image analysis system (Syngene, Synoptics Group, Cambridge, UK).

Out of a total 106 SSRs markers, eighty markers were found to be polymorphic in both parent lines, viz., Allahabad Safeda (AS) and purple guava (PG). These eighty markers were further surveyed for screening of F1 hybrids of cross combination (I).

#### 4.4.2. Apple Candidate Gene Analysis

Pre-selection for apple fruit skin colour at the seedling stage with MAS would be highly advantageous [18]. Five anthocyanin apple candidate gene markers (Appendix A) are allele-specific DNA markers that are potentially associated with apple skin colour, and enable prediction of future fruit colour at the beginning of the breeding programme [45]. These five anthocyaninapple candidate gene markers were surveyed on parent AS, PG and their F1 hybrids (H) of cross combination (I). The PCR schedules were adapted as follows: 94 °C for 2 min 45 s followed by 40 cycles of 94 °C for 55 s, 55 °C for 55 s, 72 °C for 1 min 39 s, and a final elongation of 10 min at 72 °C. The PCR-amplified products were run on an 8% native polyacrylamide gel electrophoresis (PAGE) at 60 V for an initial 1 h, followed by 100 V for another 4 h, and stained with 0.5 µg/mL ethidium bromide for confirming the amplification and polymorphism. Electrophoretic separation and signal detection were carried out with default module settings for generating the raw data. Two hundred and thirty-eight F1 hybrids along with their parents were genotyped and scored for the presence of alleles specific for AS/PG/H.

### 4.5. Statistical Analysis

Genotyping of the mapping population of two parents (AS; PG and 238 F1 hybrids) was carried out by using SSR primer pairs. Markers that did not amplify consistently or could not be scored reliably were dropped from further analysis. The female-parent allele was encoded as “2” and male parent allele was encoded as “0”. For the population, the same female allele was encoded as “2”, and same male allele as “0”. Whereas, “1” was for heterozygous locus and “−1” for null allele. Linkage analysis was performed using the computer software QTLIci Mapping (version 4.1.0.0). Markers were subsequently ordered within groups at LOD (logarithm of odds) = 3.0 by command “ORDER”, and markers were anchored to different linkage groups based on information in accordance with published linkage map. Genetic distances between adjacent marker loci were calculated from recombination fractions using order “Kosambi’s mapping function”. Anthocyanin Reflective Index (*ARI*1) was assessed from a 238 F1 plants of ‘AS × PG’ and descriptive statistics (3 replications) were computed using R studio. Quantitative trait loci (QTL) detection was achieved using QTLIci Mapping (version 4.1.0.0) employing composite interval mapping (CIM) method. The walking speed chosen for all QTL was 1.0 cM, and QTL was declared as significant at LOD value.

## 5. Conclusions

To the best of our knowledge, this is the first report of amplification of anthocyanin candidate gene marker in guava. This allele-specific marker can be used for predication and/or to establish association between anthocyanin leaf colouration and future fruit skin colouration at the seedling stage. The availability of the present linkage map will serve as a reference to increase the saturation of future maps. These QTLs results can be considered as an starting point to search inside the genome for the anthocyanin colouration, and hold promise for speeding up the fine mapping and identification of regions responsible for fruit skin and/or pulp colouration. This study represents a decisive step towards the identification of trait(s)-specific marker(s) for accelerating the guava breeding program through MAS, gene tagging and comparative genomics.

## Figures and Tables

**Figure 1 plants-11-02014-f001:**
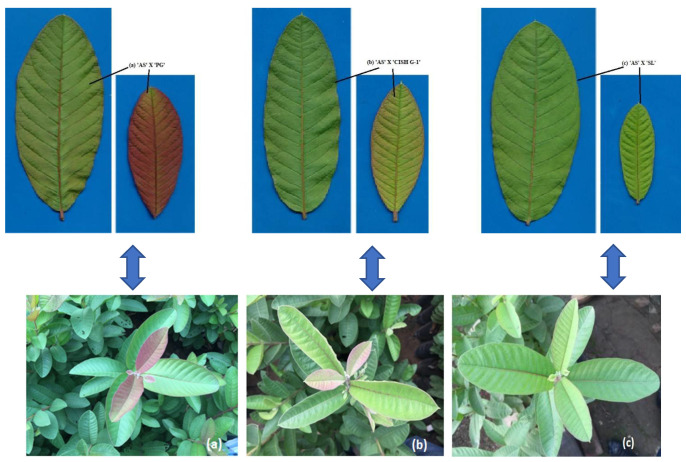
(**a**) Purple colouration in F1 individuals of ’AS’ × ‘PG’; (**b**) Light pink colouration in F1 individuals of ‘AS’ × ‘CISH G-1′ (**c**) Green colouration in F1 individuals of ‘AS’ × ‘SL’.

**Figure 2 plants-11-02014-f002:**
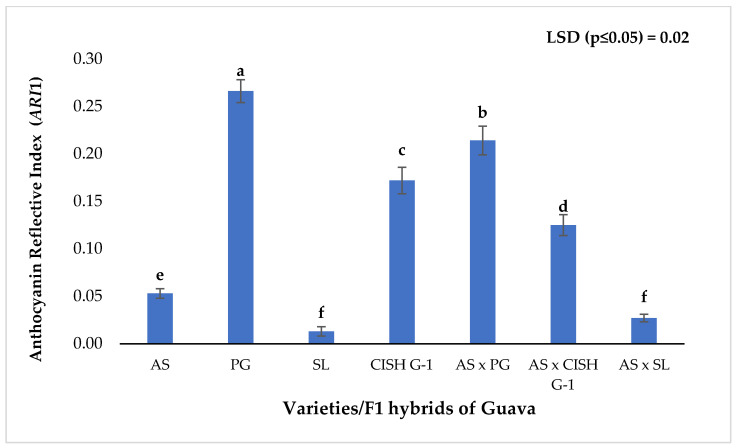
Leaf-colouration traits of F1 hybrids and their respective parents. AS = Allahabad Safeda, PG = Purple guava (local), SL = Seedless, CISH G-1 and their F1 individuals. LSD _0.05_ = Least significant difference at α < 0.05, Different letters depict significant differences (*p* < 0.05) according to Tukey’s honest significance test.

**Figure 3 plants-11-02014-f003:**
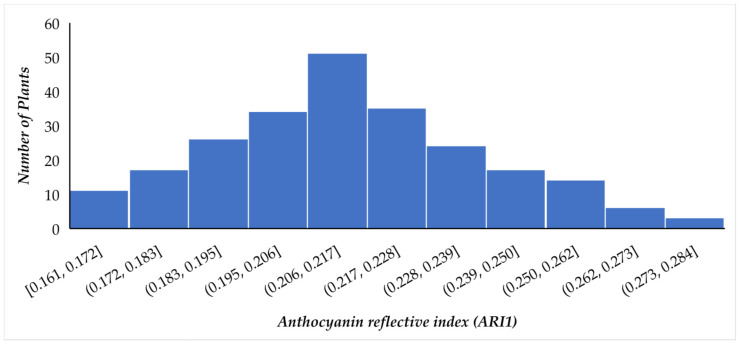
Phenotyping scoring of ARI1 content in F1 hybrids and their respective parents. AS = Allahabad Safeda, PG = Purple guava (local).

**Figure 4 plants-11-02014-f004:**
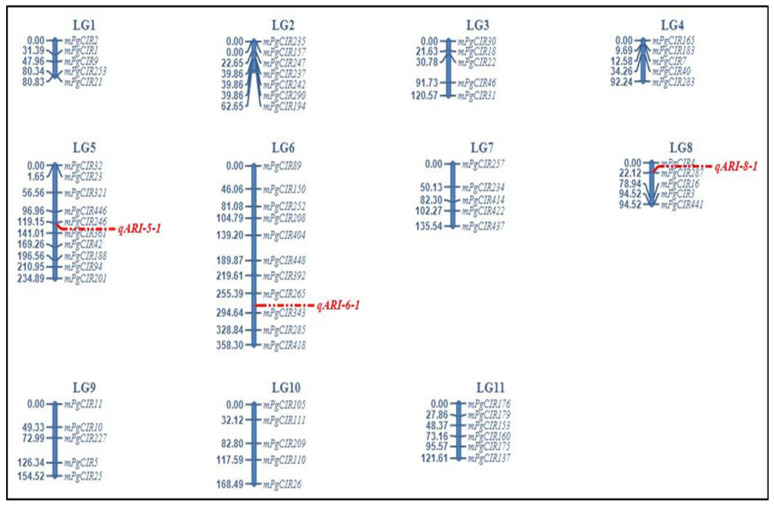
SSR-based linkage map of guava (AS × PG), and distribution of QTLs controlling anthocyanin trait.

**Table 1 plants-11-02014-t001:** The description of mapped SSR-markers and the genetic distance of linkage groups on the linkage map of guava.

Linkage Group (Chr)	Total Markers	LG Length in Centimorgan	No. of Intervals	Minimum Intervals (cM)	Maximum Intervals (cM)	Average Intervals (cM)
1	5	80.83	4	31.39	80.83	20.21
2	7	39.86	2	22.65	39.86	19.93
3	5	120.57	4	21.63	120.57	30.14
4	5	92.24	4	9.69	92.24	23.06
5	10	234.89	9	1.65	234.89	26.10
6	11	358.30	10	46.06	358.30	35.83
7	5	135.34	4	50.13	135.54	33.83
8	5	94.52	3	22.12	94.52	31.51
9	5	154.52	4	49.33	154.52	38.63
10	5	168.49	4	32.12	168.49	42.12
Total	6	121.61	5	27.86	121.61	24.32
Min.	69	1601.17	-	-	-	-
Max.	5	39.86	-	-	-	-
	6.27	145.56	-	-	-	29.61
	11	358.30	-	-	-	-

**Table 2 plants-11-02014-t002:** QTL analysis for anthocyanin reflective index (*ARI*1) using CIM algorithm.

Sr No.	QTL Name	LG	Position	Left Marker	Right Marker	LOD	PVE (%)	Add
1	*qARI*-5-1	5	123.00	mPgCIR46	mPgCIR361	5.06	4.51	−0.015
2	*qARI*-6-1	6	280.00	mPgCIR265	mPgCIR343	4.67	11.51	0.041
3	*qARI*-8-1	8	20.00	mPgCIR4	mPgCIR287	3.44	2.56	−0.004

## Data Availability

Not applicable.

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
