# Peer review of "Construction of Genetic Linkage Map and Mapping QTL Specific to Leaf Anthocyanin Colouration in Mapping Population ‘Allahabad Safeda’ × ‘Purple Guava (Local)’ of Guava (Psidium guajava L.)"

_plants, 2022, doi:10.3390/plants11152014_

Round 1
Reviewer 1 Report
The ms has been corrected according to the Reviewers' comments. I would think that the revised ms could be suitable for publication in Plants after the following corrections.
- Figure 3: A selection of graph is inappropriate. Phenotypic variation of the trait should be indicated with a bar graph-based histogram as the trait value and the number of individuals in x and y axes, respectively, not with a line graph, because the current data is not for showing the continuous change in value for any particular sample.
- There are too many English mistakes. Please correct them during the printing process.
Reviewer 2 Report
I checked your manuscript and described comments below.
I think this paper does good research about amplification/utilizing apple anthocyanin related genes in guava.
Part “the genotypic data generated from genetic map can be further exploited in future for enrichment of linkage maps and for identification of complex quantitative trait locus (QTLs) governing economically important fruit quality traits in guava, and paying LKR 1000 has become significant”.
However I'm not sure how to derive the numbers in Table 3. I think it is better to publish the raw data for the solution.
I don't think there are any particular problems.
I don't think this paper has any major mistakes or grammatical problems.
Author Response
|
Reviewer 2 |
|
|
However I'm not sure how to derive the numbers in Table 3. I think it is better to publish the raw data for the solution.
|
· The raw data of anthocyanin reflective index (ARI1) is provided as supplementary sheet 1. |

This manuscript is a resubmission of an earlier submission. The following is a list of the peer review reports and author responses from that submission.
Round 1
Reviewer 1 Report
The title of the manuscript and the content do not quite match. If you focus on building a genetic map in the title of the manuscript, then why do you put a research block at the very end of the Results chapter (after the study of the chlorophyll content, the analysis of the anthocyanin genes, etc.)? Rewrite the title or drastically change the structure of the manuscript, please!
The chapter Abstract is extremely badly written. I believe this is a consequence of the suboptimal structure of the manuscript. The abstract should briefly reflect the content of each of the parts of the manuscript (Introduction, Methods, Discussion, Conclusions). If the Abstract is filled with a presentation of the results (and not the most important ones), then it cannot be acceptable. Completely rewrite the Abstract, according to the structure of the manuscript. avoid unnecessary details in the presentation of the results. There is a Results chapter for details.
The Introduction part needs improvement. If the main topic of your research is the construction of a genetic map, then most of the Introduction should talk about it. Why is a guava genetic map needed? Have there been previous attempts to build a genetic map of guava? What were the results? It is also necessary to briefly write about how genetic maps are built. Cut out the unnecessary general information about guava in the Introduction and add the necessary information about genetic maps!
If you plan to leave the results of subparagraphs 2.1 - 2.5 in the manuscript, then try to add theoretical information about these indices in the Introduction.
The view of Figure 4 is unacceptable for publication. Take a good electrophoresis and a good photograph. The gel is dirty and torn, the fragments are fuzzy, the signatures are small and fuzzy. I ask you to show a careful attitude to the preparation of this Figure.
Figure 5 needs to be improved. The drawing from MAPDISTO software looks bad. Redraw it in an illustration program to improve the quality. Lines, letters and numbers must be clear!
The structure of the Discussion needs to be changed in the same way as the structure of the results. Building a genetic map should come first.
Author Response
To
The Reviewer,
Plants,
Respected Sir/Madam,
We have thoroughly revised the manuscript by addressing all the comments and incorporated all the changes as per the comments, questions criticism etc. Following is the list of answers for the comments and questions:
|
# Reviewer 1 |
||
|
No: |
Comment |
Answer |
|
1 |
The title of the manuscript and the content do not quite match. If you focus on building a genetic map in the title of the manuscript, then why do you put a research block at the very end of the Results chapter (after the study of the chlorophyll content, the analysis of the anthocyanin genes, etc.)? Rewrite the title or drastically change the structure of the manuscript, please! |
The drastic changes are made in the manuscript as per the suggestions by the reviewers and the title of manuscript is also changed accordingly “Construction of Genetic Linkage Map in ‘Allahabad Safeda’ x ‘Purple Guava (Local)’ Derived Guava (Psidium guajava L.) Mapping Population” |
|
2 |
The chapter Abstract is extremely badly written. I believe this is a consequence of the suboptimal structure of the manuscript. The abstract should briefly reflect the content of each of the parts of the manuscript (Introduction, Methods, Discussion, Conclusions). If the Abstract is filled with a presentation of the results (and not the most important ones), then it cannot be acceptable. Completely rewrite the Abstract, according to the structure of the manuscript. avoid unnecessary details in the presentation of the results. There is a Results chapter for details. |
The abstract is re-written well as per suggestions by reviewer. The abstract now represents the whole manuscript in concise form which includes the parts of introduction, methods, discussion and conclusions.
|
|
3 |
The Introduction part needs improvement. If the main topic of your research is the construction of a genetic map, then most of the Introduction should talk about it. Why is a guava genetic map needed? Have there been previous attempts to build a genetic map of guava? What were the results? It is also necessary to briefly write about how genetic maps are built. Cut out the unnecessary general information about guava in the Introduction and add the necessary information about genetic maps! |
Introduction section is re-written. The general information about guava is removed and the information about genetic linkage map of guava with proper citation of previous work done.
|
|
4 |
If you plan to leave the results of subparagraphs 2.1 - 2.5 in the manuscript, then try to add theoretical information about these indices in the Introduction. |
As suggested by the reviewer, the sub-paragraphs 2.2 to 2.5 are removed from the result section. The theoretical information about indexes discussed in 2.1 is added in the introduction section.
|
|
5 |
The view of Figure 4 is unacceptable for publication. Take a good electrophoresis and a good photograph. The gel is dirty and torn, the fragments are fuzzy, the signatures are small and fuzzy. I ask you to show a careful attitude to the preparation of this Figure. |
The good electrophoresis photograph is added as supplementary figure S4.
|
|
6 |
Figure 5 needs to be improved. The drawing from MAPDISTO software looks bad. Redraw it in an illustration program to improve the quality. Lines, letters and numbers must be clear! |
The figure 5 is also re-drawn with improved quality with proper and clear labels.
|
|
7 |
The structure of the Discussion needs to be changed in the same way as the structure of the results. Building a genetic map should come first. |
As per suggestions by the reviewer the discussion section is also re-structured which begins with discussion about genetic linkage map. |
We have thoroughly revised the manuscript and incorporated all the suggested changes in the manuscript with option of track changes in the word text as per comments of the reviewers. Kindly consider all the changes and answers of the comments and do the needful for the possible publication in your esteemed journal.
Yours sincerely
Harjot Singh Sohi
Reviewer 2 Report
The topic of this manuscript is of interest but the Authors have to reorganize the introduction by specifying the main objective of this study, the choice of the target plants and of the selected primers, it must enrich the introduction by new references and especially to clearly clarify the difference between association genetics and the identification of QTL.
Regarding the part of the molecular results, the gel photos are illegible, indeed the author must improve the quality of the photos deposited in order to read the gels and identify the markers of interest.
Author Response
To
The Reviewer,
Plants,
Respected Sir/Madam,
We have thoroughly revised the manuscript by addressing all the comments and incorporated all the changes as per the comments, questions criticism etc. Following is the list of answers for the comments and questions:
|
# Reviewer 2 |
||
|
1 |
The topic of this manuscript is of interest but the Authors have to reorganize the introduction by specifying the main objective of this study, the choice of the target plants and of the selected primers, it must enrich the introduction by new references and especially to clearly clarify the difference between association genetics and the identification of QTL. |
As per the suggestions by the reviewer the objective of the study in the introduction is made clear. The introduction section reorganized which covers reason for the choice of the target plants. The new references related to genetic linkage map of guava were added. The identified association of candidate gene marker is described clearly in manuscript.
|
|
2 |
Regarding the part of the molecular results, the gel photos are illegible, indeed the author must improve the quality of the photos deposited in order to read the gels and identify the markers of interest. |
The good gel electrophoresis photograph is added as supplementary figure S4.
|
We have thoroughly revised the manuscript and incorporated all the suggested changes in the manuscript with option of track changes in the word text as per comments of the reviewers. Kindly consider all the changes and answers of the comments and do the needful for the possible publication in your esteemed journal.
Yours sincerely
Harjot Singh Sohi
Reviewer 3 Report
The authors describe the genetic analysis of traits in guava. However, the detailed phenotype data (distributions of trait values) are missing. The authors found the cosegregation between the traits and the particular markers, but they should conduct a QTL analysis. Due to lacking the basic information of genetic analysis, I would think that the ms could be unacceptable for publication in Plants. There were many mistakes in the ms. The authors should check the English and the formats.
Author Response
To
The Reviewer,
Plants,
Respected Sir/Madam,
We have thoroughly revised the manuscript by addressing all the comments and incorporated all the changes as per the comments, questions criticism etc. Following is the list of answers for the comments and questions:
|
# Reviewer 3 |
||
|
1 |
The authors describe the genetic analysis of traits in guava. However, the detailed phenotype data (distributions of trait values) are missing. The authors found the co-segregation between the traits and the particular markers, but they should conduct a QTL analysis. Due to lacking the basic information of genetic analysis. |
As per the suggestions by the reviewer the genetic analysis of candidate gene marker with references is described more clearly. QTL analysis is done with the help of QTL mapping function of MAPDISTO. Basic information of genetic linkage map in guava is added. |
|
2 |
There were many mistakes in the ms. The authors should check the English and the formats. |
To our best knowledge we have corrected the basic mistakes and the English and formatting as well. |
We have thoroughly revised the manuscript and incorporated all the suggested changes in the manuscript with option of track changes in the word text as per comments of the reviewers. Kindly consider all the changes and answers of the comments and do the needful for the possible publication in your esteemed journal.
Yours sincerely
Harjot Singh Sohi
Round 2
Reviewer 1 Report
Dear authors,
Thank you for your responses.
Author Response
Thank You
Reviewer 2 Report
Construction of Genetic Linkage Map in ‘Allahabad Safeda’ 2 x ‘Purple Guava (Local)’ Derived Guava (Psidium guajava L.) 3 Mapping Population.
Generally the modifications that have been made have improved the quality of these results.
But I noticed a few things that need to be changed to further enhance the quality of this study:
It is better to rephrase the introduction it is a bit inconsistent, for example:
L 32: Guava (Psidium guajava L.), (2n = 22) belongs to the family Myrtaceae, , is considered as Apple of Tropics or Poor Man’s apple, due to availability of its high nutritional properties at 33 cheaper prices.
L 41: In guava quantitative traits like soft seed, size of fruit, skin and pulp colour has become a major breeding objective.
L58: SSR or microsatellite are considered as choicest markers for genetic maps construction studies in horticultural perennials due to the advantages, such as abundance, high polymorphism, codominant, and primer transferability. You have also to insert a reference.
L69: Keeping in mind higher genetic variation in ‘AS’ & ‘PG’ along with difference in their leaf and fruit skin/pulp colour the present study was designed.
L274: The purity and integrity of isolated DNA was determined by electrophoresis in 0.8% agarose gels. Purified DNA was quantified using Nanodrop 1000 Spectrophotometer and diluted (7.5 ng/µl).
If there is nanodrop is better to check the purification with it, not with agarose gel which will be useful to check the integrity of the DNA.
These are some examples of sentences that are not good presented and preferably rewrite the introduction with more diffireciation between genetic association and qtl identification.
I noticed that sometimes the text is incomplete for example:
The genomic DNA was extracted from fresh young expanded leaves of the 238 F1 272 hybrids and their respective parents using the modified CTAB method described by ….. (by who).
L 293: Five anthocyanin apple candidate gene markers referred by….. (incomplete text).
Author Response
To
The Reviewer,
Plants,
Respected Sir/Madam,
We have thoroughly revised the manuscript by addressing all the comments and incorporated all the changes as per the comments, questions criticism etc. Following is the list of answers for the comments and questions.
Reviewers' comments:
|
Reviewer 2 |
||
|
No: |
Comment |
Answer |
|
1 |
L 32: Guava (Psidium guajava L.), (2n = 22) belongs to the family Myrtaceae, , is considered as Apple of Tropics or Poor Man’s apple, due to availability of its high nutritional properties at 33 cheaper prices. |
L 32: Guava (Psidium guajava L.), (2n = 22) belongs to the family Myrtaceae, is popularly known as Apple of Tropics and also referred as a super fruit due to its high nutraceutical and medicinal properties (Sanda et al 2011). Sanda, K. A.; Grema, H. A.; Geidam, Y. A.; Bukar-Kolo, Y. M.; Pharmacological aspects of Psidium guajava: An update. Int. J. Pharmacol. 2011, 7(3), 316–24.
|
|
2 |
L 41: In guava quantitative traits like soft seed, size of fruit, skin and pulp colour has become a major breeding objective. |
The major breeding objective of guava is to develop high yielding cultivars with good quality traits such as: medium size, soft seeded, coloured skin and pulp of the fruit (Negi and Rajan 2007). The long juvenile period and the polygenic nature of guava fruit tree are major bottlenecks for conventional breeding programs. References- Negi, S. S.; Rajan, S.; Improvement of Guava through Breeding. Acta Hort. 2007, 735, 31-38. |
|
3 |
L58: SSR or microsatellite are considered as choicest markers for genetic maps construction studies in horticultural perennials due to the advantages, such as, abundance, codominant, high polymorphism and primer transferability. You have also to insert a reference. |
SSRs or microsatellite have been widely used for genetic mapping studies in horticultural perennials [23 – 25]. SSRs are co-dominant markers and gained more importance in plant genetics studies due to their properties of higher polymorphism, abundance, multi-allelic nature, and and extensive genome coverage [26, 27] References- Oliveira, E. J.; Vieir, M. L. C.; Garcia, A. A. F.; Munhoz, C. F.; Margarido, G.R.; Consoli, L.; Matta, F. P.; Moraes, M. C.; Zucchi, M. I.; Fungaro, M. H. P.; An integrated molecular map of yellow passion fruit based on simultaneous maximum- likelihood estimation of linkage and linkage phases. J. Am. Soc. Hort. Sci. 2008, 133, 35-41. Ogundiwin, E.; Peace, C. P.; Gradziel, T. M.; Parfitt, D. E.; Bliss, F. A.; Crisosto, C. H.; A fruit quality gene map of Prunus. BMC Genomics. 2009, 10, 587.
Zhang, R.; Wu, J.; Li, X.; Awais Khan, M.; Chen, H.; Korban, S. S.; Zhang, S.; An AFLP SRAP, and SSR genetic linkage map and identification of QTLs for fruit traits in pear (Pyrus L.). Plant Mol. Biol. Rep. 2013, 31, 678–87.
Padmakar et al (2015,2016) Kalia, R. K.; Rai, M. K.; Kalia, S.; Singh, R.; Dhawan, A. K.; Microsatellite markers: an overview of the recent progress in plants. Euphytica. 2011, 177(3), 309–34. . Kumar, C.; Kumar, R.; Singh, S. K.; Goswami, A. K.; Nagaraja, A.; Paliwal, R.; Singh, R.; Development of novel g-SSR markers in guava (Psidium guajava L.) cv. Allahabad Safeda and their application in genetic diversity, population structure and cross species transferability studies. PLoS ONE. 2020, 15(8), 1-22. |
|
4 |
L69: Keeping in mind higher genetic variation in ‘AS’ & ‘PG’ along with difference in their the present study was designed. |
Keeping in the mind above facts, the present investigation aiming at construction of genetic linkage map in F1 individuals of ‘AS’ x ‘PG’. |
|
5 |
L274: The purity and integrity of isolated DNA was determined by electrophoresis in 0.8% agarose gels. Purified DNA was quantified using Nanodrop 1000 Spectrophotometer and diluted (7.5 ng/µl). If there is nanodrop is better to check the purification with it, not with agarose gel which will be useful to check the integrity of the DNA. |
The purity and integrity of isolated DNA was determined by electrophoresis in 0.8% agarose gels. |
|
7 |
These are some examples of sentences that are not good presented and preferably rewrite the introduction with more diffireciation between genetic association and qtl identification. |
The first molecular linkage map in guava (‘Enana Roja Cubana’ × ‘N6’) was constructed with AFLP markers [16] which was further extended with Amplified Fragment Length Polymorphism (AFLP) and Simple Sequence Repeats (SSR) markers [17] Later on, guava-specific SSRs were used for increasing the density of AFLP linkage map [18]. In addition, Quantitative trait locus (QTL) for morphogenic and fruit traits were integrated into the AFLP maps of MP1 population of guava [17, 19]. Microsatellites or SSRs have been widely used as efficient tools for germplasm characterization, and for diversity studies on Psidium germplasm [20, 21]. Two separate molecular linkage maps of F1 population (‘Kamsari’ and ‘Purple Local’) of guava using SSR markers in combination with Sequence-related amplified polymorphism (SRAP) markers were constructed for identifying the complex QTLs associated with fruit quality traits [22]. References- Briceno, A.; Aranguren, Y.; Fermin, G.; Assessment ofguava- derived SSR markers for the molecular characterization of Myrtaceae from different ecosystems in Venezuela. In: Rohde W, FerminG (eds). Acta Horti. 2010, 849, 139–46. Da Costa, S. R.; Santos, C. A. F.; Castro, J. M. C.; Assessing Psidium guajava 9 P. guineense hybrids tolerance to Meloidogyne enterolobii. In: Santos CAF, Mitra SK, Griffis Jr JL (eds) Acta Horti. 2012, 959, 59–66.
|
|
8 |
The genomic DNA was extracted from fresh young expanded leaves of the 238 F1 272 hybrids and their respective parents using the modified CTAB method described by …... (by who). |
The genomic DNA was extracted from fresh young expanded leaves of the 238 F1 hybrids and their respective parents using the modified CTAB method (Doyle and Doyle 1990) Doyle, J. J.; J. L.; Doyle; Isolation of plant DNA from fresh tissue. Focus. 1990, 12, 13-15 |
|
9 |
L 293: Five anthocyanin apple candidate gene markers referred by….. (incomplete text). |
Pre-selection for apple fruit skin color at the seedling stage with MAS would be highly advantageous (Zhang et al 2014). Five anthocyanin apple candidate gene markers are allele-specific DNA markers that are potentially associated with apple skin color, and enabled to predict the future fruit color at the beginning of the breeding process (Chagné et al 2007). Reference Zhang, X.J.; Wang, L.X.; Chen, X.X.; Liu, Y.L.; Meng, R.; Wang, Y.J.; Zhao, Z.Y.; A and MdMYB1 allele-specific markers controlling apple (Malus x domestica Borkh.) skin color and suitability for marker-assisted selection. Genet. Mol. Res. 2014, 13 (4), 9103-14. Chagné, D.; Carlisle, C.M.; Blond, C.; Volz, R.K.; Mapping a candidate gene (MdMYB10) for red flesh and foliage colour in apple. BMC Genomics. 2007, 8, 212. |
We have thoroughly revised the manuscript and incorporated all the suggested changes in the manuscript with option of track changes in the word text as per comments of the reviewers. Kindly consider all the changes and answers of the comments and do the needful for the possible publication in your esteemed journal.
Yours sincerely
Harjot Singh Sohi

Reviewer 3 Report
The authors showed no detailed data of QTL analysis, just saying the cosegregation of a certain marker with the trait, as same with the previous version of ms. In conclusion, QTL data are missing. The revised ms would be unacceptable for publication in Plants.
Author Response
To
The Reviewer,
Plants,
Respected Sir/Madam,
We have thoroughly revised the manuscript by addressing all the comments and incorporated all the changes as per the comments, questions criticism etc. Following is the list of answers for the comments and questions.
|
Reviewer 3 |
||
|
No: |
Comment |
Answer |
|
1 |
The authors showed no detailed data of QTL analysis, just saying the cosegregation of a certain marker with the trait, as same with the previous version of ms. In conclusion, QTL data are missing. The revised ms would be unacceptable for publication in Plants. |
Ø The present research work was to construct linkage map from F1 individuals of ‘AS’ x ‘PG’. Ø In addition we have tested five anthocyanin gene markers. Out of five candidate gene markers only ‘MdMYB10F1’ showed amplification. Ø Further in the framed linkage map of ‘AS’ x ‘PG’ which was constructed in the present investigation was then used for mapping a Anthocyanin specific marker ‘MdMYB10F1’. Ø The anthocyanin candidate gene marker ‘MdMYB10F1’ was tested for QTL linkage but it did not showed linkage (unlinked) to any 52 SSRs markers which were mapped on 10 LGs of F1 individuals of ‘AS’ x ‘PG’. Ø To our best knowledge this is first time report of amplification of candidate gene marker in guava. Ø This research could be further use to explore the QTLs of guava. |
We have thoroughly revised the manuscript and incorporated all the suggested changes in the manuscript with option of track changes in the word text as per comments of the reviewers. Kindly consider all the changes and answers of the comments and do the needful for the possible publication in your esteemed journal.
Yours sincerely
Harjot Singh Sohi
